# Amyloid Properties of the FXR1 Protein Are Conserved in Evolution of Vertebrates

**DOI:** 10.3390/ijms23147997

**Published:** 2022-07-20

**Authors:** Maria E. Velizhanina, Alexey P. Galkin

**Affiliations:** 1Laboratory of Amyloid Biology, St. Petersburg State University, Universitetskaya Emb. 7/9, 199034 St. Petersburg, Russia; velizhanina.me@gmail.com; 2Laboratory of Signal Regulation, All-Russia Research Institute for Agricultural Microbiology, Podbelskogo 3, 196608 St. Petersburg, Russia; 3Department of Genetics and Biotechnology, St. Petersburg State University, Universitetskaya Emb. 7/9, 199034 St. Petersburg, Russia; 4St. Petersburg Branch, Vavilov Institute of General Genetics, Universitetskaya Emb. 7/9, 199034 St. Petersburg, Russia

**Keywords:** functional amyloid, FXR1 protein, brain, evolution, vertebrates, fish, amphibians, reptiles, birds, mammals

## Abstract

Functional amyloids are fibrillary proteins with a cross-β structure that play a structural or regulatory role in pro- and eukaryotes. Previously, we have demonstrated that the RNA-binding FXR1 protein functions in an amyloid form in the rat brain. This RNA-binding protein plays an important role in the regulation of long-term memory, emotions, and cancer. Here, we evaluate the amyloid properties of FXR1 in organisms representing various classes of vertebrates. We show the colocalization of FXR1 with amyloid-specific dyes in the neurons of amphibians, reptiles, and birds. Moreover, FXR1, as with other amyloids, forms detergent-resistant insoluble aggregates in all studied animals. The FXR1 protein isolated by immunoprecipitation from the brains of different vertebrate species forms fibrils, which show yellow-green birefringence after staining with Congo red. Our data indicate that in the evolution of vertebrates, FXR1 acquired amyloid properties at least 365 million years ago. Based on the obtained data, we discuss the possible role of FXR1 amyloid fibrils in the regulation of vital processes in the brain of vertebrates.

## 1. Introduction

Amyloids are protein fibrils with an ordered cross-β structure. Protein monomers form β layers and are arranged parallel or antiparallel to each other and perpendicular to the fibril axis. The term “amyloid” was traditionally used to describe extracellular pathological tissue deposits of protein fibrils connected to systemic and neurodegenerative diseases. However, many studies have reported that proteins capable of forming intracellular and extracellular fibrils with a cross-β structure do not always cause pathological outcomes but can regulate vital processes in bacteria and eukaryotes [1,2,3,4,5,6]. Such proteins are commonly referred to as functional amyloids [7]. Amyloid fibrils have unique properties, such as unprecedented resistance to a variety of agents, mechanical strength, and elasticity [8]. In this regard, it is unsurprising that eukaryotes evolved to employ amyloid structures in various vital processes. For example, it is known that a fragment of the PMEL17 protein forms amyloid fibrils, which are necessary for the synthesis and polymerization of melanin [1]. Another example of functional amyloids is the Orb2 protein, which forms amyloid granules that interact with RNA molecules in fruit fly neurons [6]. We have also shown that the RNA-binding protein FXR1 functions in the rat brain and human neuronal cell culture in the amyloid form [5]. All these examples support the concept of functional amyloids, which postulates that the ability of a protein to form amyloid fibrils may confer adaptive advantages, and this property may be fixed in the course of evolution [7,9].

In this work, we decided to assess at what stage of evolution the amyloid fibrils of the FXR1 protein became a constitutive component of brain neurons. This protein belongs to the Fragile X-Related (FXR) family of RNA-binding proteins, which also includes Fragile X Mental Retardation (FMRP) and Fragile X-Related 2 (FXR2) proteins. FXR1 and FXR2, unlike FMRP, arose at the dawn of the vertebrates’ evolution [10]. FXR1 is expressed in different cell types in higher eukaryotes, including muscle cells and neurons. This protein contains RNA-binding motives (KH1/KH2 and RGG) and differentially regulates the stability and translation of various RNA molecules such as miR301a-3p, p21, cMYC, microRNA-9, microRNA-124, and TNF-α [11,12,13,14,15]. Interaction with RNA molecules determines the critical role of FXR1 in regulating long-term memory, emotions, and cancer [12,14,16,17].

Previously, we have demonstrated that FXR1 colocalizes with amyloid-specific dyes CR (congo red), and Thioflavin S and T in cortical neurons of the rat brain. FXR1 extracted from the brain by immunoprecipitation shows yellow-green birefringence after staining with CR. Moreover, FXR1 in the rat brain is presented as detergent-resistant oligomers and insoluble aggregates only. RNA molecules that are colocalized with FXR1 amyloid particles in cortical neurons are insensitive to treatment with RNase A [5]. The N-terminal amyloid-forming region of FXR1 is highly conserved in mammals. The sequence of FXR1 (1–379 aa) in the rat and mouse is identical. It differs from human, macaque, and cat FXR1N sequences only by one amino acid [5].

Here, using a set of cytological, immunological, and biochemical approaches, we evaluated the amyloid properties of FXR1 in the brains of amphibians, reptiles, and birds. The obtained data allowed to assess the evolutionary conservatism of the amyloid properties of the FXR1 protein in vertebrates.

## 2. Results

### 2.1. FXR1 Forms SDS-Resistant Amyloid-like Aggregates in the Brains of Amphibians, Reptiles, and Birds

All amyloids characterized up until now are resistant to treatment with ionic detergents and, in particular, to SDS (sodium dodecyl sulphate). We have previously shown that FXR1 forms aggregates in the rat brain that are resistant to treatment with 1% SDS at room temperature [5]. To assess the SDS-resistance of the FXR1 aggregates in the brain of vertebrates, we used commercially available antibodies ab129089 (Abcam, Cambridge, UK). According to the manufacturer’s protocol, these antibodies specifically recognize the human, mouse, and rat FXR1 protein. Our control experiments have shown that these antibodies specifically bind the FXR1 protein of all major groups of jawed vertebrates, except fish. Obviously, this is determined by the fact that the FXR1 protein is highly conserved among vertebrates. The experiments were carried out using brain samples from clawed frogs (*Xenopus laevis*), red-eared turtles (*Trachemys scripta*), and domestic chickens (*Gallus gallus domesticus*). To check whether FXR1 forms SDS-resistant aggregates in vivo, total protein lysate from the brain was treated with 1% SDS at room temperature and separated into three fractions: (1) proteins less than 100 kDa; (2) oligomers larger than 100 kDa; (3) insoluble aggregates. Using an ab129089 antibody, we demonstrated that the major fraction of FXR1 is detected as insoluble aggregates in all studied vertebrate species (Figure 1). A small proportion of the protein is found in the oligomer fraction only in the chicken brain. The monomeric form of the protein is not identified by visual inspection in any of the studied organisms. The experiments were repeated three times. The distribution of protein by fractions is shown in diagrams (Figure 1). The obtained results indicate that the FXR1 protein, as with the previously characterized amyloids, forms SDS-resistant aggregates in the brain of all studied terrestrial vertebrate species.

### 2.2. FXR1 Colocalizes with Amyloid-Specific Dyes Thioflavin S and Congo Red on Cryosections of the Brain of Clawed Frog, Red-Eared Turtle, and Chicken

Amyloid fibrils of various proteins bind specifically to Thioflavin S and CR dyes both in vivo and in vitro. To check whether FXR1 is present in an amyloid form in the brain, we compared the localization of FXR1 with the localization of amyloid-specific dyes on cryosections of the brain of frogs, red-eared turtles, and chickens. The FXR1 protein is observed in the perinuclear cytoplasm of neurons in all analyzed animal species (Figure 2A–C). These results are consistent with previously published data on the localization of FXR1 in mouse, rat, and human cells [5,18]. Thioflavin S (green signal) definitively colocalizes with FXR1 antibodies (red signal). Thioflavin S brightly stains individual granules in the area of FXR1 localization in the neurons of turtles. However, in chickens and frogs, Thioflavin S stains the site of protein localization more evenly. Most likely, the bright signal of Thioflavin S corresponds to large and compact aggregates of the studied protein. Colocalization of Thioflavin S and FXR1 was estimated using Pearson’s coefficient for 100 random zones, shown as mean ± SEM (Appendix A). Pearson’s correlation coefficient in domestic chicken, red-eared turtle, and clawed frog neurons is 0.78, 0.70, and 0.69, respectively.

It should be noted that Thioflavin S and its analog Thioflavin T, in addition to amyloid fibrils, can effectively bind to some other protein structures [19,20]. In order to test the amyloid properties of FXR1, we also assessed its colocalization with CR in brain neurons of various vertebrate species (Figure 3A–C). CR signals (red) are detected in the area of FXR1 localization (green) in brain neurons. CR, as with Thioflavin S, most brightly stains large and compact granules in turtle neurons (Figure 3B). The colocalization of CR and FXR1 was estimated using Pearson’s coefficient for 100 random zones, shown as mean ± SEM (Appendix A). Pearson’s correlation coefficient for the colocalization of CR and FXR1 in domestic chicken, red-eared turtle, and clawed frog neurons was 0.68, 0.53, and 0.66, respectively.

In polarized light, no apple-green birefringence is detected after staining with CR on cryosections of brain samples from clawed frogs, red-eared turtles, and chickens. A similar observation was made previously, in experiments with the rat brain, where we showed that the yellow-green polarization was detected only after immunoprecipitation of FXR1, but not on cytological preparations [5]. In all likelihood, CR is a highly specific but not sensitive enough dye for the cytological detection of birefringence of intracellular amyloids.

### 2.3. FXR1 Isolated from the Clawed Frog, Red-Eared Turtle, and Chicken Brain Is Detected as Fibrils That Stain with CR and Show Yellow-Green Birefringence

FXR1 was extracted from the brain of frogs, red-eared turtles, and chickens using the approach that we described earlier [5]. The protein lysates were incubated with ab129089 anti-FXR1 antibodies immobilized onto magnetic beads. Then, immunoprecipitated material was washed, eluted without boiling, and concentrated by centrifugation. This method makes it possible to isolate native, nondenatured fibrils [5,21]. The results presented in Figure 4 show that FXR1 isolated from different animals forms fibrils that are detected by TEM (transmission electron microscopy). These fibrils bind CR and demonstrate yellow-green birefringence (Figure 4). Considering that FXR1 forms SDS-resistant aggregates, colocalizes with amyloid-specific dyes on cryosections, and shows yellow-green birefringence after immunoprecipitation and staining with CR, we conclude that this protein is present in amyloid form in the brains of amphibians, reptiles, and birds.

### 2.4. Functional Amyloid Fibrils Arose in the Brain at the Dawn of the Evolution of Jawed Vertebrates

We have characterized the amyloid properties of the FXR1 protein in neurons of the brain representatives of all classes of jawed vertebrates excluding fish. Commercially available antibodies do not detect this protein in fish brains. As an alternative approach, we used conformation-dependent OC antibodies that allow cytological detection of amyloid fibrils of various proteins, such as Aβ peptide, islet amyloid deposits, and Orb2 [22,23]. The sections of the brain of the 5XFAD transgenic mice were used as a positive control. Amyloid plaques of Aβ are detected in the brain of these mice as early as 6 months of age [24]. The OC antibodies recognize not only intercellular Aβ deposits, but also intracellular structures around the nuclei of cortical neurons of the 5XFAD transgenic mice (Appendix A). Moreover, anti-amyloid OC antibodies recognize perinuclear structures in brain neurons of all studied vertebrate species, including fish (Figure 5). These structures have the same localization as the FXR1 protein. Unfortunately, it is impossible to use rabbit ab129089 anti-FXR1 antibodies and rabbit OC antibodies simultaneously for co-localization experiments. However, these results indicate that amyloid-like structures exist in brain neurons of all groups of jawed vertebrates.

As already mentioned, the N-terminal amyloidogenic sequence of FXR1 (1–379 aa) is highly conserved among mammals. We conducted a comparative analysis of this sequence in all vertebrate species of this study. The sequence of the rat protein was used as a reference. The percentages of the FXR1 (1–379 aa) sequence identity of zebrafish, clawed frogs, red-eared turtles, and chickens compared to *Rattus norvegicus* are 82.3%, 92.1%, 97.1%, and 96.6%, respectively (Figure 6). These data argue that the N-terminal amyloidogenic region of FXR1 is highly conserved among vertebrates. Using bioinformatics algorithm ArchCandy [25], we showed that the N-terminal fragment of FXR1 contains two potentially amyloidogenic sequences that are located identically in all the studied animals (Figure 6). Another bioinformatics algorithm (MetAmyl) [26] predicts the presence of many short amyloidogenic sequences in an N-terminal fragment of FXR1 that are largely similar in localization in different vertebrate species (Appendix A).

## 3. Discussion

We have shown that the FXR1 protein is present in the brains of amphibians, reptiles, and birds in the amyloid form. Previously, the same results were obtained in the study of this protein in the rat brain and in culture of human neuroblastoma cells [5]. This protein is detected in the brain of vertebrates in the form of SDS-resistant aggregates and colocalizes in neurons with amyloid-specific dyes (see Figure 1, Figure 2 and Figure 3). In addition, the FXR1 protein isolated by immunoprecipitation from the brains of all studied animals is detected as fibrils that stain with CR and exhibit a yellow-green birefringence, a classic characteristic of amyloids. Ex vivo, we observe large fibrils, but it cannot be ruled out that small amyloid structures combine into large fibrils during the process of their isolation from the brain. Amyloid fibrils tend to interact laterally [27]. Following immunoprecipitation, we condense the protein by centrifugation and the amyloid particles can combine into large bundles.

We demonstrated that amyloid fibrils of FXR1 became an integral component of neurons during vertebrate evolution, at least in amphibians. This is the first case of amyloids identification in reptiles and birds. Our data also suggest that the FXR1 protein can be presented in the amyloid form in the fish brain. This protein belongs to the X-Related (FXR) family, which also includes FMRP and FXR2 proteins. A search using the NCBI protein database showed that FMRP is an ancient protein originating in invertebrates such as cnidarians, worms, and insects. Proteins FXR1 and FXR2 are orthologs of FMRP, which appeared only in vertebrates. All these proteins bind RNA and have (KH) domains and an RGG box [28]. These three proteins have common as well as unique RNA targets [11,29]. We have previously shown that RNA in FXR1-containing RNP particles in rat brain neurons are very stable and insensitive to RNase A treatment [5]. What is the biological relevance of forming stable RNP particles containing the amyloid protein? Thus far, we can only put forward a hypothesis based on mammalian cell culture studies. In experiments with cell culture HEK293, when cells are grown in the presence of serum, FXR1 forms large RNP granules that bind the mRNA of TNFα and prevent its translation. In the absence of serum, large RNP granules are disassembled, and FXR1 activates the translation of TNFα mRNA [30]. According to published data, FXR1 binds not only to TNFα RNA molecules but also to several other RNAs such as miR301a-3p, p21, cMYC, microRNA-9, and microRNA-124 [11,12,13,14,15] involved in the regulation of cellular response to stress. It is quite possible that, under stress, large amyloid FXR1-containing particles are disassembled in brain neurons. Such dynamic changes in RNP granules may contribute to the activation of RNA molecules involved in regulating of the stress response. Our data suggest that such an adaptive mechanism could have been fixed in evolution with the appearance of amphibians, and perhaps even earlier.

The N-terminal sequence of FXR1 (1–379 aa) amino acids is responsible for its aggregation in the rat brain [5]. Here, we show that this fragment of FXR1 is highly conserved in vertebrates (Figure 5). Interestingly, despite the amino acid substitutions, the potentially amyloidogenic regions of FXR1 are located at the same position in the protein sequence in all studied vertebrate species (Figure 5). We believe that this cannot be a coincidence, and the formation of amyloid RNP particles in vertebrate brain neurons is an adaptive characteristic.

In conclusion, we have shown that the amyloid properties of the RNA-binding FXR1 protein are evolutionarily conserved in jawed vertebrates. This protein is found in the amyloid form in brain neurons of evolutionarily distant groups of vertebrates such as amphibians, reptiles, birds, and mammals. Based on our data and published studies, we discuss the hypothesis that the FXR1-containing RNP particles allow the storage of RNA molecules that play a crucial role in the stress response. Our data indicate that amyloid fibrils during vertebrate evolution became an integral component of brain neurons at least 365 million years ago.

## 4. Materials and Methods

### 4.1. Animals

Domestic chickens (*Gallus gallus domesticus*) of the Russian White breed, aged 6–12 months, were purchased from the All-Russian Research Institute of Genetics and Farm Animal Breeding “Genofond” (Pushkin, Leningrad Region, Russia). Mature adults of *Trachemys scripta*, *Xenopus laevis,* and *Danio rerio* were purchased from breeders. The study was conducted according to the guidelines of the Declaration of Helsinki and was approved by the Institutional Ethics Committee of St. Petersburg State University (Statement #131-03-6 issued 01.06.2017).

### 4.2. Preparation of the Brain Cryosections and Brain Homogenization

For immunohistochemistry, brains of selected animals were dissected, washed with PBS, and fixed in 4% PFA for 3 h. After rinsing with PBS, brain samples were embedded in FSC22 compound (Leica, Wetzlar, Germany), frozen in liquid nitrogen, and stored at −70 °C. Cryosections of 20 µm thick were made using cryostat CM1850UV (Leica, Wetzlar, Germany). The prepared sections were mounted on glass slides, air-dried, and stored at −20 °C.

For protein fractionation and immunoprecipitation, the brains of the studied animal species were frozen in liquid nitrogen immediately after dissection. The obtained samples were homogenized using a cryogenic laboratory mill Freezer/Mill 6870 (SPEX SamplePrep, St. Metuchen, NJ, USA).

### 4.3. Protein Analysis

Protein extraction from brains was performed as follows: 100 mg of brain homogenate was resuspended in 500 µL of PBS buffer with protease inhibitors Complete Mini Tablets (Roche, Mannheim, Germany). Then, SDS solution was added to the samples to the final concentration of 1% and incubated for 15 min at room temperature. Cell debris was removed by low-speed centrifugation (805× g, 5 min). Then, the lysate was ultracentrifuged for 1 h 45 min (258,500× g, +18 °C) and the insoluble fraction was collected. The supernatant was loaded into the Amicon Ultra 100 K filter unit (Merck Millipore, Burlington, MA, USA) and further divided by centrifugation into two fractions: less than 100 kDa (monomeric fraction) and higher than 100 kDa (oligomeric fraction). The volume of all three fractions was equalized. The FXR1 protein was detected with the primary anti-FXR1 antibody ab129089 (Abcam, Cambridge, UK) and secondary Goat Anti-Rabbit IgG H&L conjugated with HRP (ab205718) (Abcam, Cambridge, UK). Chemiluminescent detection was performed using the Amersham ECL Prime Western Blotting Detection Reagent (GE Healthcare, Chicago, IL, USA). The intensity of the protein bands was calculated using the program Image Lab (Bio-Rad, Hercules, CA, USA).

### 4.4. Immunoprecipitation

The extraction of the native FXR1 protein was performed as a two-step procedure. First, anti-FXR1 antibodies ab129089 (Abcam, Cambridge, UK) were coupled to SureBeads magnetic beads coated with protein A (Bio-Rad, Hercules, CA, USA). For this, beads were prewashed with a binding buffer (BB) (PBS, 0.02% Tween-2) and then resuspended in 500 μL of BB with the addition of Complete Protease Inhibitor Cocktail (Roche, Mannheim, Germany). An amount of 5 μL of anti-FXR1 antibodies was added to the beads (final concentration of antibodies, 6.43 μg/mL), and the mixture was incubated at room temperature for 1 h under slow overhead rotation. After the incubation, the magnetic beads were washed with the BB. Next, brain lysates were incubated with the prepared magnetic beads overnight at +4 °C under slow overhead rotation. Then, the beads were washed with BB to remove unbound proteins, resuspended in the elution buffer (125 mM Glycine, 0.1% Triton-X100, pH 2.1), and incubated for 10 min to ensure protein elution. Then, an aliquot of 1.5 M Tris pH 8.8 was added to the eluate to restore the pH. Samples were concentrated by centrifugation at 436,000× *g* for 2 h at +4 °C. The resulting precipitate was dissolved in 10 μL of cold sterile water. Fibril formation was verified by TEM and CR staining.

### 4.5. Electron Microscopy

Negative-stained samples were prepared on formvar-coated copper grids (FCF300-Cu-50) (EMS, Hatfield, PA, USA). A 10 μL aliquot of the suspension after immunoprecipitation was adsorbed to the formvar film for 1 min, washed twice with 10 μL of water for 10 s, stained with 10 μL of 1% uranyl acetate for 1 min, and air-dried. TEM images were obtained using a Jeol JEM-2100 microscope.

### 4.6. CR Staining and Polarization Microscopy

For polarization microscopy, the protein samples after immunoprecipitation were resuspended in 10 μL of water, spotted on a glass slide, and air-dried. Preparations were stained with 0.25% water solution of CR for 5 min at room temperature. The samples were washed with water and covered with a cover slip. Brightfield and polarization microscopy images were acquired using a polarizing microscope BIOLAR PI PZO equipped with a camera ToupCam UCMOS10000KPA and ToupView(x86) software.

### 4.7. Immunohistochemistry and Staining with Amyloid-Specific Dyes

Conformation-dependent amyloid-specific OC antibodies (AB2286, Merck, Burlington, MA, USA) were used at a dilution of 1:400, and the primary antibodies anti-FXR1 (ab129089, Abcam, Cambridge, UK) were used at a dilution of 1:600. Cryosections were pretreated as described earlier [5] and incubated with primary antibodies at +4 °C, overnight. After washing to remove unbound antibodies, the sections were incubated with anti-rabbit secondary antibodies conjugated with Alexa Fluor 647 (ab150075, Abcam, Cambridge, UK) at a dilution of 1:1000 for an hour at +37 °C. To visualize nuclei, the slides were counterstained with DAPI. Histological staining of amyloids was performed using 0.1 mg/mL CR solution in 50% ethanol or using 1% Thioflavin S solution in 70% ethanol. Visualization of the results of immunohistochemical analysis and staining with amyloid-specific dyes was performed with a TCS SP5 confocal microscope (Leica Microsystems, Wetzlar, Germany) and “Leica Application Suite X 3.3.0.16799” software. To quantify the colocalization of FXR1 with CR red or Thioflavin S, Pearson’s correlation coefficient was calculated using the coloc2 plugin of FIJI software (http://fiji.sc/Fiji, accessed on 23 February 2020). At least 100 zones on preparations of each animal studied were analyzed to assess the colocalization of CR or Thioflavin S with the FXR1 protein.

### 4.8. Comparative Analysis of Sequences of FXR1 Protein in Different Vertebrate Species

For comparative analysis of the N-terminal region of the FXR1 protein of different vertebrate species, the following sequences were taken from the NCBI protein database: XP_038958508.1 (isoform X1, *Rattus norvegicus*); XP_046779863.1 (isoform X1, *Gallus gallus domesticus*); XP_034636516.1 (isoform X1, *Trachemys scripta*); NP_001081786.1 (homolog A, *Xenopus laevis*); NP_001299810.1 (isoform 1, *Danio rerio*). The sequence of *Rattus norvegicus* was chosen as a reference. Potentially amyloidogenic sequences were identified by the algorithm ArchCandy (Ahmed et al., 2015).

## Figures and Tables

**Figure 1 ijms-23-07997-f001:**
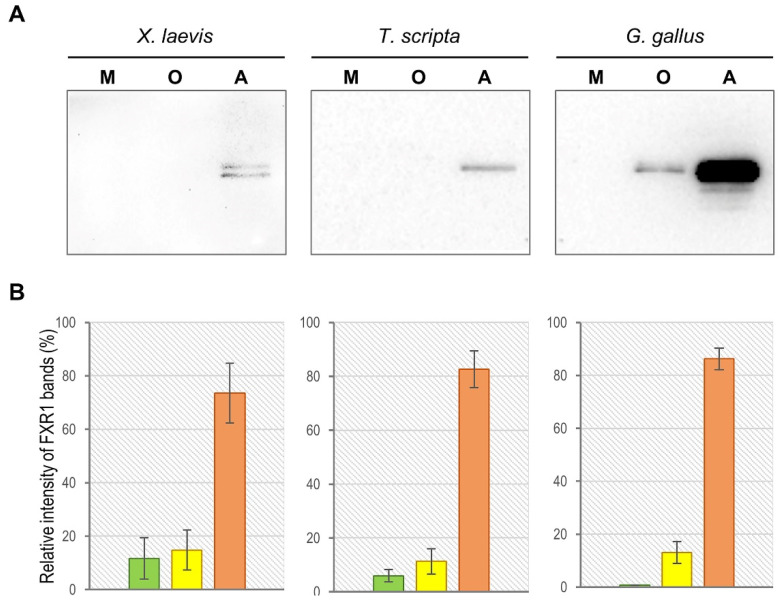
FXR1 forms SDS-resistant amyloid-like aggregates in the brains of *Xenopus laevis*, *Trachemys scripta,* and *Gallus gallus domesticus*. (**A**) FXR1 is predominantly found as insoluble aggregates in all vertebrate species studied. Total protein lysate from the brain was treated with 1% SDS and separated into three fractions: of monomers less than 100 kDa (M), oligomers larger than 100 kDa (O), and insoluble aggregates (A). The fractions were subjected to SDS-PAGE and analyzed by immunoblotting with anti-FXR1 antibodies. (**B**) Relative intensity of bands corresponding to FXR1 monomers, oligomers, and insoluble aggregates is represented as mean ± SEM (standard error of the mean) for three independent brain samples for each organism. The green, yellow, and orange columns of the diagram represent the values for monomers, oligomers, and insoluble aggregates, respectively.

**Figure 2 ijms-23-07997-f002:**
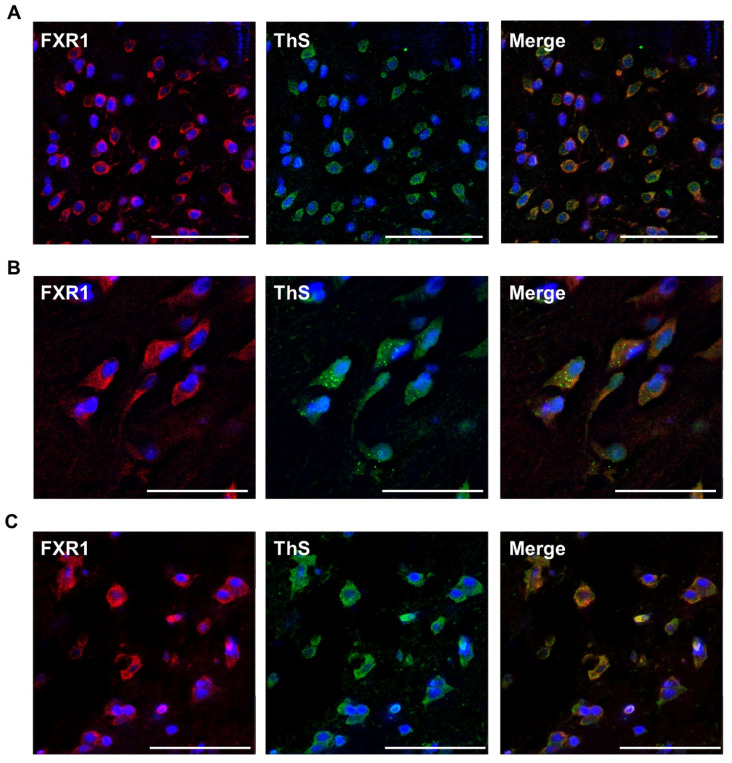
Colocalization of FXR1 (red signal) with amyloid-specific dye Thioflavin S (green signal) in neurons of frog (**A**), turtle (**B**), and chicken (**C**) brain sections. Blue signal corresponds to the nuclear dye DAPI; scale bar is 50 µm.

**Figure 3 ijms-23-07997-f003:**
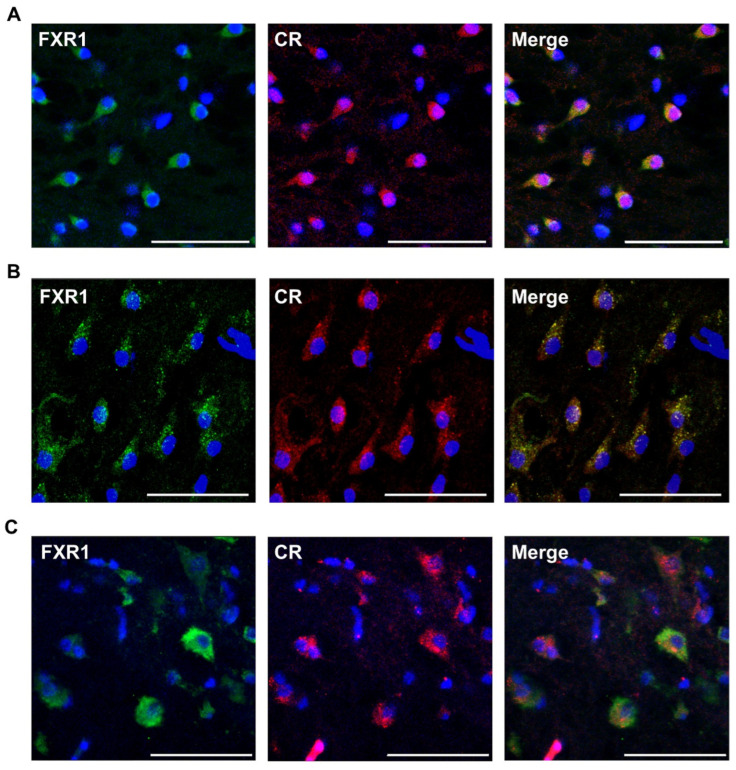
Colocalization of FXR1 (green signal) with amyloid-specific dye CR (red signal) in neurons of frog (**A**), turtle (**B**), and chicken (**C**). Blue signal corresponds to the nuclear dye DAPI; scale bar is 50 µm.

**Figure 4 ijms-23-07997-f004:**
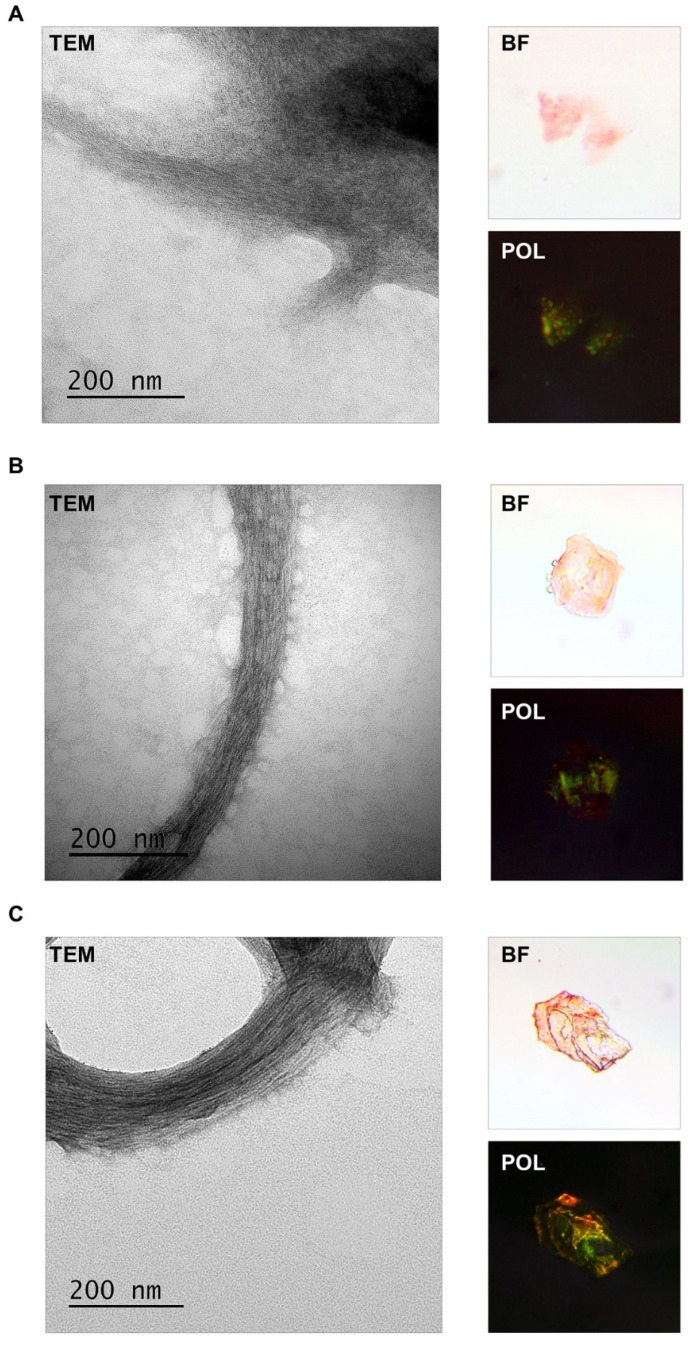
Analysis of the fibrillar structure and CR staining of the FXR1 protein isolated from the brain of *Xenopus laevis* (**A**), *Trachemys scripta* (**B**), and *Gallus gallus domesticus* (**C**). FXR1 after immunoprecipitation was detected using TEM. CR staining was analyzed in transmitted (BF) and polarized (POL) light.

**Figure 5 ijms-23-07997-f005:**
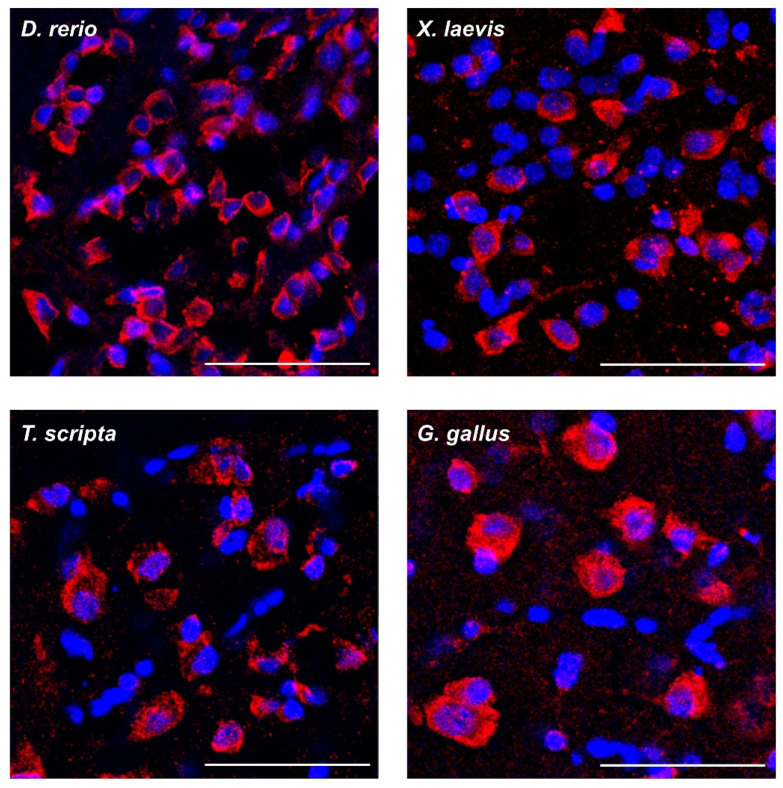
Anti-amyloid OC antibodies (red signal) recognize cytoplasmic structures in brain neurons of all studied vertebrate species. Blue signal corresponds to the nuclear dye DAPI; scale bar is 50 µm.

**Figure 6 ijms-23-07997-f006:**
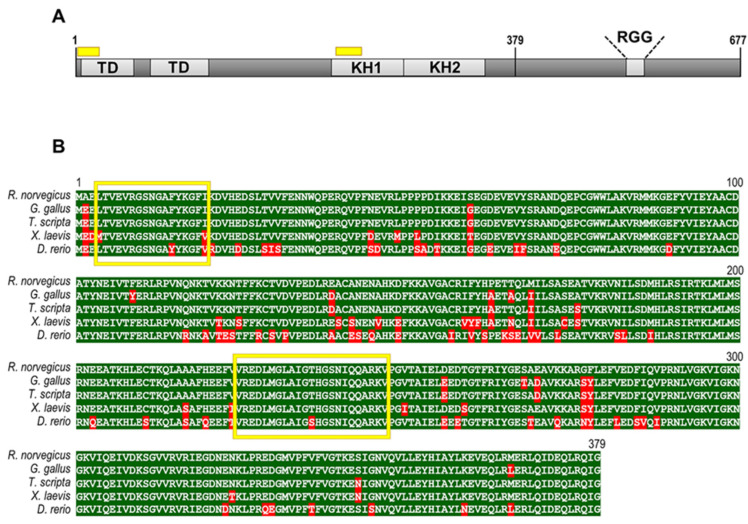
(**A**) Schematic representation of the FXR1 protein structure of *Rattus norvegicus*. The potentially amyloidogenic regions are indicated by yellow boxes. TD—Tudor domains responsible for the recognition of trimethylated peptides; KH1, KH2, and RGG—RNA-binding domains. (**B**) Comparative analysis of sequences of N-terminal amyloidogenic region of the FXR1 protein in *Rattus norvegicus*, *Gallus gallus domesticus*, *Trachemys scripta*, *Xenopus laevis*, and *Danio rerio*. The sequence of Rattus norvegicus was chosen as a reference. Red color indicates amino acid residues that differ from the reference sequence. The yellow frame indicates potentially amyloidogenic sequences identified by the algorithm ArchCandy.

## Data Availability

Not applicable.

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
