# Peer review of "Amyloid Properties of the FXR1 Protein Are Conserved in Evolution of Vertebrates"

_ijms, 2022, doi:10.3390/ijms23147997_

Round 1
Reviewer 1 Report
The authors address the hypothesis of functional amyloid role of the fragile X-related protein 0 (FXR1).
After their previous study concerning rat, mouse and human FXR1 (ref. 5), they nnow extend the study to frog, turtle and chicken, to confirm the early emergence of functional amyloid properties of FXR1 in the tree of life.
They first show that FXR1 is a high molecular weight (>100 kDa) stable aggregate when in brain lysate (sect. 2.1).
They find colocalization of FXR1 in amyloid form with ThT, ThS, and CR stain (sect. 2.2).
They observe the isolated FXR1 as amyloid in TEM (sect. 2.3).
Conclusion at lines 161-162 seems adequately supported.
Amyloid aggregation after isolation is mentioned (lines 213-215) as a possible bias.
The high conservation of N-terminal sequence among different species is reported (sect. 2.4).
The hypothesis of an RNA dynamic storage/release associated to FXR1/RNA assemblies is shortly discussed.
In summary, the manuscript adequately supports the presence of FXR1 amyloid aggregates in the central nervous system of different species.
Since FXR is a protein family associated to mental disorders, this observation, as others reported in the literature, points to pathology associated to the lack of scaffolding rather than to amyloid deposits by themselves.
I find this observation important and worth to be published in IJMS.
Minor points are described below.
The sequence of FXR1 should be introduced.
For instance what fraction of FXR1 is the discussed N-terminus (379 residues).
When possible, acronyms should be introduced the first time they are used:
L.61 - CR (Congo red).
L.77 - SDS (sodium dodecyl sulphate)
L.106 - SEM.
etc.
Author Response
Thank you for your attention to our manuscript.
- The sequence of FXR1 should be introduced. For instance what fraction of FXR1 is the discussed N-terminus (379 residues).
We have added a picture with a schematic representation of the FXR1 protein sequence (Fig. 6A).
- When possible, acronyms should be introduced the first time they are used:
L.61 - CR (Congo red).
L.77 - SDS (sodium dodecyl sulphate)
L.106 – SEM.
etc.
In response to the remark, we have inserted the decoding of all abbreviations in the text of the article.
Reviewer 2 Report
The authors previously demonstrated that the protein FXR1 forms amyloid in rat brain and human neuronal cells in culture. In this study, they investigated whether FXR1 also functions in amyloid form in other terrestrial vertebrate species. For this purpose, they used brain samples from clawed frog, red-eared turtle, and domestic chicken using cytological, immunological, and biochemical approaches.
They revealed that FXR1 of these three vertebrates forms a SDS-resistant insoluble aggregates, which is a common feature of amyloid. FXR1 was found to colocalize with Thioflavin S and Congo Red in the brain cryosections of the used animals. They also checked the morphology using TEM and confirmed that FXR1 takes on fibrillar aggregates. From these observations, the authors concluded that FXR1 forms amyloids in brains of reptiles and birds as well as amphibians and mammalians. Together with other findings, they speculate that the amyloid formation of FXR1 is relevant to stress response and regulation of translation.
The present finding will expand the knowledge of amyloid fibrils in terms of biological functions in different animal species. I think this manuscript will be accepted after minor modifications as follows.
My only concern is:
In Fig. 6, I recommend that the authors test the amyloidgenicity of the N-terminal region using MetAmyl (http://metamyl.genouest.org/e107_plugins/metamyl_aggregation/db_prediction_meta.php) because the results are sometimes software-dependent.
Minor points:
l.61 The term "CR" appears here for the first time. It should be written as "Congo Red (CR)".
l.160 "immunopreciptation", not "immunorecipitation".
l.255 What is "during of"?
Author Response
Thank you for your attention to our manuscript.
- In Fig. 6, I recommend that the authors test the amyloidgenicity of the N-terminal region using MetAmyl (http://metamyl.genouest.org/e107_plugins/metamyl_aggregation/db_prediction_meta.php) because the results are sometimes software-dependent.
We analysed the presence of potentially amyloidogenic sequences using the MetAmyl bioinformatics algorithm. This algorithm predicts the presence of many short amyloidogenic sequences that are largely similar in localization in different vertebrate species. These data have been added to the manuscript:
“Another bioinformatics algorithm (MetAmyl) [26] predicts the presence of many short amyloidogenic sequences in N-terminal fragment of FXR1 that are largely similar in localization in different vertebrate species (Supplementary Figures 3)”.
- 61 The term "CR" appears here for the first time. It should be written as "Congo Red (CR)".
Correction done.
- 160 "immunopreciptation", not "immunorecipitation".
Correction done.
- 255 What is "during of"?
Correction done. The phrase "during of evolution" has been corrected to "during evolution”.